# Effects of Polybrominated Diphenyl Ethers on Hormonal and Reproductive Health in E-Waste-Exposed Population: A Systematic Review

**DOI:** 10.3390/ijerph19137820

**Published:** 2022-06-25

**Authors:** Vishal Singh, Javier Cortes-Ramirez, Leisa-Maree Toms, Thilakshika Sooriyagoda, Shamshad Karatela

**Affiliations:** 1School of Public Health and Social Work, Queensland University of Technology, Brisbane, QLD 4059, Australia; leisamaree.toms@qut.edu.au (L.-M.T.); thilakshikadanushi.sooriyagoda@connect.qut.edu.au (T.S.); 2Centre for Data Science, Queensland University of Technology, Brisbane, QLD 4059, Australia; javier.cortesramirez@qut.edu.au; 3Children’s Health and Environment Program, The University of Queensland, Brisbane, QLD 4101, Australia; 4Faculty of Medical and Health Sciences, Universidad de Santander, Cúcuta 540003, Colombia; 5School of Pharmacy, University of Queensland, Brisbane, QLD 4072, Australia; s.karatela@uq.edu.au; 6Australian Institute of Tropical Health and Medicine, James Cook University, Mackay, QLD 4740, Australia

**Keywords:** flame retardants, persistent organic pollutants (POPs), endocrine-disrupting chemicals, environmental exposure, occupational health, non-monotonic dose–response

## Abstract

Electronic waste management is a global rising concern that is primarily being handled by informal recycling practices. These release a mix of potentially hazardous chemicals, which is an important public health concern. These chemicals include polybrominated diphenyl ethers (PBDEs), used as flame retardants in electronic parts, which are persistent in nature and show bioaccumulative characteristics. Although PBDEs are suspected endocrine disruptors, particularly targeting thyroid and reproductive hormone functions, the relationship of PBDEs with these health effects are not well established. We used the Navigation Guide methodology to conduct a systematic review of studies in populations exposed to e-waste to better understand the relationships of these persistent flame retardants with hormonal and reproductive health. We assessed nineteen studies that fit our pre-determined inclusion criteria for risk of bias, indirectness, inconsistency, imprecision, and other criteria that helped rate the overall evidence for its quality and strength of evidence. The studies suggest PBDEs may have an adverse effect on thyroid hormones, reproductive hormones, semen quality, and neonatal health. However, more research is required to establish a relationship of these effects in the e-waste-exposed population. We identified the limitations of the data available and made recommendations for future scientific work.

## 1. Introduction

Electronic waste (e-waste) has become the fastest-growing source of waste worldwide due to the increasing electronic device consumption and obsolescence caused by the rapid advancements in technology, lack of backwards compatibility, unserviceable parts, and high cost of repairs [1,2]. Less than 20% of e-waste is formally recycled, and the rest is either abandoned in the environment, buried in landfills, or recycled informally [3]. In 2019, 53.6 million metric tons (Mt) of e-waste was generated worldwide, a rise of 9.2% from five years ago, and this is expected to reach 74 Mt by 2030 [3]. The number of people informally engaged in the e-waste recycling sector is not known, but estimates suggest the number to be in the millions, of which approximately a quarter are women, with a large proportion in their childbearing years [4]. Informal e-waste recycling has potential impacts on vulnerable populations, including recycling workers and nearby residents, due to the chemicals released in the environment from dismantling, open-burning, and other processing methods conducted in an uncontrolled manner.

Chemicals released in e-waste recycling processes include lead, cadmium, lithium, polycyclic aromatic hydrocarbons, dioxins, and especially flame retardants (FRs), depending on the composition and temperatures involved in processing [5,6]. Flame retardants are compounds used as additives in polymers, rubbers, fabrics, and other substances for preventing, delaying, or slowing down the spread of fire chemically and/or physically [7,8,9]. High concentrations of FRs have been found near e-waste recycling sites in air, dust, soil, human milk, crops, poultry, and water, thus posing an exposure risk to humans through multiple routes [10,11,12,13,14,15]. The most common types of FRs found in e-waste recycling sites are brominated flame retardants (BFRs), of which polybrominated diphenyl ethers (PBDEs) are legacy BFRs that have been being phased out by industries since the mid-2000s voluntarily and due to regulations, except for decabromodiphenyl ether [9,16]. Yet, they are abundant in the environment because of their heavy use in the past and their persistent nature [17,18]. There are 209 congeners of PBDEs (generally written as BDE-1 to BDE-209), of which many are used in electronics (Figure 1) [19]. These compounds are usually available as mixtures of multiple PBDE congeners that are released when electronics containing these mixtures as flame retardants are burnt [20]. BFRs are generally concentrated in soil and sediments rather than in water as they are lipophilic and have low solubility in water and low vapour pressure [21]. Many PBDEs are highly persistent (such as BDE-153 with a half-life of around 6.5 years in the human body), causing bioaccumulation and biomagnification and abetting long-distance travel and reaching multiple ecosystems, even the Arctic ecosystem [20,22,23,24].

In a process known as debromination, higher-brominated PBDE congeners can be broken down into lower-brominated congeners that are more mobile and toxic [22,25]. Consequently, debromination lengthens their persistence in the environment and increases their spread into different ecosystems as lower-brominated PBDE congeners have higher solubility in water and are more bioaccumulative [26]. Due to these properties of PBDEs, humans are at risk of direct exposure through products containing PBDEs and indirectly through other sources, including e-waste recycling.

Brominated FRs can enter the human body via ingestion, inhalation, dermal contact, and transplacental transfer from mother to foetus [27,28]. The dominant pathway of exposure in e-waste recycling areas is potentially through dietary exposure for residents closely followed by dust ingestion and air inhalation, especially for workers, who also experience higher exposure contribution through skin contact [29]. In vitro and in silico studies have shown that PBDEs and their metabolites are endocrine disruptors, especially targeting thyroid and reproductive function due to their structural similarities with hormones, competing to bond with the hormone receptors [30,31,32]. Thyroid hormones influence multiple organ systems, including the cardiovascular, respiratory, gastrointestinal, and sympathetic nervous systems, testicular development, spermatogenesis, and foetal growth [33,34,35]. On the other hand, a balance of reproductive hormones is essential to regulate the menstrual cycle, pregnancy, fertility, foetal growth, spermatogenesis, and secondary sexual characteristics [33]. Thus, potential alterations in hormones caused by PBDEs could have health implications on the exposed individual and their future offspring.

Despite the potential health effects associated with PBDE exposure, few studies have reviewed the hormonal health implications of these agents in e-waste-exposed populations. These populations are distinct relatives to the general population as they are exposed to higher doses of PBDEs. Therefore, studies focusing on e-waste-exposed populations are pivotal in understanding the behaviour of PBDEs at high doses. This paper aims to identify and analyse research studies that address the hormonal and reproductive health effects of PBDEs in e-waste-exposed populations.

## 2. Methods

A review protocol was prepared following the Navigation Guide methodology for systematic reviews [36]. This tool was used because it is designed to assess data from clinical science for use in environmental health research and is based on the Cochrane Collaboration and Grading of Recommendations Assessment Development and Evaluation (GRADE) [37]. The population, exposure, reference group, and outcome (PECO) were defined as follows:◯Population: Nearby residents of e-waste recycling sites or workers;◯Exposure: Polybrominated diphenyl ethers (PBDEs);◯Reference group: Non-exposed population or population with minimal risk of exposure to e-waste;◯Outcome: Hormonal and/or reproductive health effects observed due to exposure to PBDEs through e-waste.

### 2.1. Search Strategy, Study Selection and Data Extraction

Searches in PubMed, Embase, and Scopus were conducted using combinations of the keywords: flame retardant, polybrominated diphenyl ether, PBDE, organohalogen, persistent organic pollutant, electronic waste, health, hormone, reproduction, development, effect, and disrupt (the search strategy is presented in Appendix A). Reference lists were assessed to identify additional studies that could be included in the review.

The following inclusion criteria were considered: studies measuring the association of at least one hormonal or reproductive health outcome with PBDE exposure levels based on biomonitoring measures; exposed populations that live near e-waste recycling facilities or are occupationally exposed to e-waste; studies of human populations. The following exclusion criteria were considered: studies only measuring exposures or potential risk rather than health outcomes; full-text not available; and articles published in languages other than English.

To compare exposure levels using a consistent dataset, median and range values were converted to mean and standard deviation using the formulae recommended by Hozo et al. [38] as per the sample sizes. If a study performed multiple statistical analyses, the outcome data adjusted for most confounders were extracted without any modifications.

### 2.2. Quality Assessment

A quality assessment of the selected papers was performed as described by Woodruff and Sutton [36]. In brief, the risk of bias was assessed in nine domains: study group representation, exposure assessment methods, outcome assessment methods, confounding, incomplete outcome data, selective outcome reporting, financial conflict of interest, and presence of any other biases. There were five possible ratings in each criterion: ‘low’, ‘probably low’, ‘probably high’, and ‘high’ risk of bias.

The overall quality of evidence was assessed as ‘low’, ‘moderate’, or ‘high’ based on five potential downgrade factors (risk of bias, indirectness, inconsistency, imprecision, and publication bias) and three upgrade factors (large magnitude of effect, dose–response, and whether the residual confounding underestimates the overall effect size). The strength of evidence across studies was determined based on the quality of the body of evidence, direction of effect, confidence in effect, and other compelling attributes, such as if the association could be attributed to other causes. The combination of all factors considered would result in ‘sufficient’, ‘limited’, ‘inadequate’, or ‘evidence of lack of toxicity’ (Appendix A). The rationale for judgment were provided for each step of the quality and strength of evidence assessment.

## 3. Results

There were 1997 records initially identified, of which 993 remained after removing duplicates. The title and abstract screening excluded 951 articles, and 42 articles were screened for full-text review. After adding an additional study found by searching the references, 19 studies were included in the review. The literature search and screening process flowchart is shown in Figure 2, and a list of included studies is provided in Appendix A. Thirteen studies (68.4%) assessed thyroid-related health outcomes [39,40,41,42,43,44,45,46,47,48,49,50,51], three studies (15.7%) assessed reproductive hormones [40,49,52], two studies (10.5%) assessed semen quality [49,53], and six studies (31.5%) measured neonatal health [44,51,54,55,56,57]. In addition, one study assessed adrenocorticotropic hormone (ACTH), growth hormone, and cortisol, and two studies assessed insulin-like growth factor 1 (IGF-1) [46,48,57]. The most frequently assessed PBDE congeners in the studies (>84%) were selected for the review: BDE-28, 47, 99, 100, 153, 154, 183, and 209. Further study descriptions are presented in Table 1, and the concentration of congeners, hormone levels, and measurement of other matrices are presented in Appendix A.

The selected studies were published from 2005 to 2020, with 15 studies (78.9%) from China, and 1 study from Vietnam [39], Canada [40], and Sweden [43], respectively, with sample sizes ranging from 14 to 442. Fourteen studies (73.6%) followed a case–control design, and four studies (21%) were cross-sectional analyses. One of the studies was a longitudinal cohort study involving exposed and unexposed phases at varying temporal and spatial points [43]. Three studies (15.8%) recruited informal e-waste recycling workers, and two studies (10.5%) recruited formal e-waste recycling workers, whereas fifteen studies (78.9%) recruited residents in e-waste-exposed areas. Participants who had any relevant diseases or took medicines related to the health effects that were to be assessed were excluded in all except seven studies (36.8%) [39,40,44,45,48,50,51]. The significant associations of health effects with PBDE congeners are presented in Table 2. The associations of health effects with PBDE congeners are presented in Appendix A.

The ratings assigned for risk of bias are shown in Table 3. More than 95% of studies had a ‘low’ or ‘probably low’ risk of bias ratings across all the domains except for confounding (79%), incomplete outcome data (79%), and selective outcome reporting (47%). Details of the risk of bias ratings are provided in Appendix A. The quality assessment and strength of evidence assessment for each health outcome assessed across the studies are presented in Table 4. The evidence available for thyroid hormones, reproductive hormones, and neonatal health was of ‘moderate quality’ and provided ‘limited evidence of toxicity’, while the evidence for semen quality was of ‘low quality’ with ‘limited evidence of toxicity’, and the evidence for other hormones (i.e., cortisol, growth hormones, ACTH, and IGF-1) was of ‘low quality’ and provided ‘inadequate evidence of toxicity’. Explanations regarding quality and strength assessment are presented in Appendix A.

## 4. Discussion

We used a rigorous method with the application of the Navigation Guide methodology to systematically assess nineteen studies to identify a positive association of PBDE congeners identified in the studies with total triiodothyronine (TT3) and negative associations with total thyroxine (TT4), free T3 (FT3), and free T4 (FT4). We also identified significant adverse effects on reproductive hormones, semen quality, and neonatal health. This study offers insights into the current state of the scientific literature and provides a structured discussion regarding the hormonal and reproductive health effects of PBDEs resulting from e-waste exposure. This is the first systematic review of research conducted on the e-waste-exposed population regarding the hormonal and reproductive health effects of PBDEs.

### 4.1. PBDE Exposure Levels

The PBDE exposure levels were generally higher amongst the included studies when compared to the general population of most countries, except for the populations of some studies in North America [19], possibly due to stringent fire protection requirements and high production rates. Pregnant female participants were an exception, potentially due to changes in behaviour to reduce exposure, for example, cleaning house dust [58] or avoiding participation in e-waste-recycling-related activities. The exposure levels were related to the occupation of participants and the temporal length of residence in the exposed area. Generally, the exposure levels were higher in occupationally exposed participants due to proximity to electronic waste and children due to behavioural factors and proportionally smaller bodyweights for similar exposure through the environment and food consumption [29]. We identified that the exposure to PBDEs among the studies reviewed was between 1.64- and 6.99-fold higher in the exposed populations than in the control populations, which concurs with a previous review on the global distribution of PBDE levels [59], demonstrating the vulnerable environment near e-waste sites.

### 4.2. Hormonal Health

Increased PBDE levels were positively correlated with TT3, while TT4 and FT4 were negatively correlated. All PBDE congeners except BDE-47 were negatively associated with FT3 levels, while thyroid-stimulating hormone (TSH) varied in the direction and magnitude of effect across the studies. However, the overall strength of evidence was not sufficiently high, as the included studies did not adjust for some confounders that may play a significant role in thyroid homeostasis, such as a family history of thyroid diseases [60], iodine levels [61], and the consideration of other potential interactions with endocrine-disrupting chemicals (EDCs) present near an e-waste recycling site. Furthermore, most of the associations were non-significant, which could mean that the effect of PBDEs on thyroid hormones is small or has a large variance and requires a larger sample size to observe the effects with statistical significance. Other reasons for the non-significant and inconsistent findings across the studies could be due to multiple modes of action of PBDEs in the human body, co-exposure to other established EDCs, and the possibility of a non-monotonic dose–response (NMDR), in which a change in the shape of the dose–response graph is observed with an increase in exposure to a contaminant [62].

Thyroid hormones exert their influence in the body at extremely low doses and are maintained through complex feedback mechanisms such that interactions of even low doses of PBDEs can cause disruptions in homeostasis through multiple known pathways [63,64,65,66]. For example, PBDEs and their metabolites can alter thyroid hormones by their interaction with deiodinases, which are enzymes that help in the synthesis of T3 from T4, by competing with T4 to bind at the active site [67]. As a consequence, PBDE exposure will inhibit T4 deiodination, which is required to form T3 (i.e., T4 will be upregulated, and T3 will be downregulated). Statistically significant T4 upregulation or T3 downregulation was not reported by any of the selected studies; however, we found two studies that exhibited these effects at lower exposure levels than populations in this review [68,69], which could mean that PBDEs do not favour this mode of action at higher dosage levels.

PBDEs can also compete with T4 to bind with thyroxin-binding globulin and transthyretin, which transport thyroid hormones and prolong their half-life, and thyroid hormone receptor alpha and thyroid hormone receptor beta, which are thyroid hormone nuclear receptors responsible for the transcription of genes [66,70,71]. This will decrease T4 levels in the blood and cause thyroid gland hyperactivity as a feedback response [72], similar to our review’s findings. An additional mechanism for a decrease in T4 levels could be due to the mixed-type inducer activity of PBDEs on hepatic phase I and phase II enzymes [73,74,75,76]. These modes of action of PBDEs follow key characteristics of EDCs [77] and show that disruptions in thyroid hormone levels are possible in either direction, which increases the plausibility of an NMDR.

Previous studies regarding the effect of PBDEs on TT4 and TSH hormones in the general population suggest that there may be an NMDR, forming a U-shaped curve where the hormones are downregulated at low doses of PBDEs and upregulated at higher doses of PBDEs [78]. An NMDR is conceivable considering the complexities of thyroid hormone secretion and feedback mechanisms, multiple PBDE modes of action, and some observational studies obeying a U-shaped curve. However, not all studies agree with the relationship suggested by Zhao et al. [78], specifically Guo et al. [41] and Guo et al. [42], who showed that the PBDE levels were high, but reported significant negative associations for TT4 and TSH with BDE- 47, 100, 153, 183, 209, and ∑PBDEs. Furthermore, Wang et al. [45] and Xu et al. [47] found that TSH levels were lower in exposed populations with high (>100 ng/g lipid) ∑PBDE levels, albeit no significant associations were reported. The contrasts observed when trying to fit the findings of this review in a U-shaped curve can be partly explained by differences in the methodologies between studies, differences in the populations selected, the presence of biases, and the simultaneous presence of other EDCs and their interactions with thyroid function. This lack of complete understanding of the PBDEs’ interaction with thyroid function emphasizes the need for additional evidence from larger-scale studies.

The associations seen in the selected studies evinced that PBDEs exert different effects in men and women regarding reproductive hormones. Testosterone levels increased in men, while there was no change in women; similarly, follicle-stimulating hormone (FSH) levels decreased in women, while there was no change in men. Significant results reported for estradiol were inconsistent in men across studies, which could not be explained by differences in exposure levels, adding to the fact that another study outside the search criteria showed dissimilar results around the same exposure levels [79]. Luteinizing hormone was positively associated in three other studies regarding male subjects, similar to the results of Guo et al. [52,79,80,81]. In two studies from other populations with male subjects [79,82], BDE-209 showed a negative relationship, while other congeners showed a positive relationship between testosterone and PBDEs, whereas studies selected in this review did not report significant associations with BDE-209 but reported a significant positive relationship with testosterone for other congeners. These inconsistent results for BDE-209 compared to other congeners regarding the relationship with testosterone could be because the congener did not show androgen-antagonistic activities in vitro, in contrast to most other congeners [31]. We only found one study outside the e-waste-exposed population that studied female reproductive hormones, which also showed a negative association of PBDEs with FSH. PBDEs are linked with lower fertility and fecundity in women [83]. Therefore, more attention is warranted in this area to prevent adverse effects.

Similar to interactions with thyroid hormones, in vitro and in silico studies show that PBDEs and their metabolites can possibly disrupt reproductive hormones through agonistic and antagonistic binding with oestrogen-related receptor (ERR) γ [84], agonistic binding with ERRα and ERRβ [85], and anti-androgenicity [86,87], indirectly increasing oestrogen bioavailability by inhibiting the formation of oestradiol glucuronidation [88], agonistic binding with progesterone, and inhibiting oestradiol [31]. PBDEs can potentially show an NMDR as they show an affinity to bind with multiple receptor pathways that may cause different binding patterns at varying exposure levels [89]. Even though there were a limited number of studies that met the inclusion criteria, studies from other populations and the demonstration of potential modes of action also add to the evidence suggesting PBDEs cause reproductive hormone imbalance; however, more research is required to understand the behaviour of PBDEs with reproductive hormones in both males and females at varying exposure levels.

### 4.3. Reproductive Health

The findings in this review suggest that PBDEs negatively affect neonates, although the number of studies is limited. We identified increased birth length, adverse birth outcomes, and decreased head circumference, birth weight, BMI, and Apgar score in neonates as the most common health problems. Thyroid hormones are possibly related to the associations observed for neonatal health matrices and PBDE exposures, as they play a significant role in the development of the foetus and survival after birth [90]. During the first two trimesters, the foetus is dependent on maternal thyroid hormones, as it does not produce its own thyroid hormones. In the third trimester, thyroid hormones aid in foetal maturation changes necessary to prepare for extrauterine life, such as respiratory and cardiovascular functions that are essential for neonatal viability [90]. PBDEs suppress placental growth, possibly by increasing the production of reactive oxygen species and DNA methylation, which can mediate a restrictive growth environment for the foetus by reducing nutrient and hormone transport, resulting in foetal-growth-restricted infants [91,92,93]. The smaller placental size coupled with altered maternal thyroid hormones as an implication of exposure to PBDEs may be responsible for the lower birth weights observed in the studies in this review and other studies involving general populations [91,94,95,96,97,98,99,100]. In addition, we identified two studies with a decline in other metrics used to estimate foetus development, such as abdominal circumference, femur length, and biparietal diameter in mothers with higher levels of PBDEs [99,101]. Head circumference is used as a proxy for estimating brain development, and a decrease in size suggests that PBDEs have a neurological effect, which concurs with previous research that found lower IQs were associated with increased PBDE levels [37]. This may stem from the adverse impact PBDEs have on thyroid function, which plays a vital role in neuropsychological development [102,103].

There were conflicting results observed for head circumference with other studies. Li et al. [104], Miranda et al. [105], and Lopez-Espinosa et al. [99] observed a decrease, while Yin et al. [106] and Chen et al. [107] observed an increase associated with PBDE exposures. Robledo Candace et al. [100] identified sex-specific differences in neonates, in which head circumference and birth weight associated with PBDEs were lower in girls and higher in boys. These disparities can be due to differences in how thyroid hormones act in women and men [108]. Other conflicting results were found when comparing our review results to previous research regarding aspects such as birth length, in which a decrease was associated with PBDE exposure [91,98,100,104]. An increase in gestational age was supported by Chen et al. [98], but Eick et al. [94] and Jin et al. [91] observed a negative association with PBDE exposure. Additionally, there were other studies that did not find statistically significant associations between neonatal health indices and PBDE exposure levels [109,110] or found associations that were incomparable to the results from our selection of studies, such as associations with hypospadias in neonates [111,112]. The conflicting results could have been due to the differences in confounder adjustments, the statistical analyses used, multiple chemical exposures, congener profiles, and the study design.

Only two studies investigated semen quality with small sample sizes, and the results were selectively reported, but strong negative associations were found for semen volume, sperm concentration, and sperm count. Previous studies have shown a negative association of sperm concentration with exposure to PBDEs in the general population [113,114,115]. On the other hand, a study observed an increase in sperm concentration and total sperm count due to exposure to PBDEs [116], while no decline in semen quality was observed in another study [117]. The level of PBDEs in the blood samples of all other population groups was lower than that in the e-waste-exposed population. Semen quality variables not included in this review also declined, such as the morphology of sperm cells with elevated PBDE levels in house dust [49] and sperm mobility and motility with an increase in PBDE biomarker levels [82,113,116,118]. An altered state of thyroid hormones can lead to semen abnormalities through their actions on cells in testis [119]. PBDEs may impair semen quality given the high magnitude of effect seen in the selected studies and similar declines observed in other population groups; however, current evidence is limited, highlighting the need for further investigation with larger sample sizes.

### 4.4. Limitations and Recommendations

Although exposure to e-waste is a concern in other regions, especially African and south-east Asian countries, most studies in this review assessed populations in China; hence, the strength of representing these demographics is limited. Furthermore, effects on children are understudied, considering they have the highest exposure following occupationally exposed people and are the most vulnerable populations to environmental exposures [120]. The articles in this review that assessed effects on children due to PBDE exposure observed significant alterations in thyroid hormones [41,48], alongside the association of prenatal maternal exposures with neonatal health indices, which could be an indication of possible long-term health implications. Therefore, longitudinal cohort studies are suggested for children exposed to PBDEs to improve our understanding of their long-term health impacts.

In the studies within this review and in e-waste areas generally, there is a mix of chemicals present in the environment, with some being more potent than others, and these chemicals are also correlated with each other in terms of the concentrations found in human matrices [46,50]. Two studies that explored interactions with other endocrine disruptors found significant interactive associations [42,52]. The probability that other EDCs, such as phenols, phthalates, polychlorinated biphenyls, dioxins, and furans, to name a few, could have exerted some influence on the observed health outcomes cannot be dismissed. Additionally, it is probable that an NMDR exists for the relationship between PBDEs and thyroid hormones, and the associations observed are exclusive to e-waste-exposed populations with high levels of PBDE exposure. Therefore, further research on PBDEs should consider these aspects in their study design.

Non-uniformity in the study design, such as in the sampling methods, confounder adjustments, and statistical analyses, reduced the comparability amongst studies. Stricter confounder adjustments need to be addressed in many studies as hormone levels depend on age, gender, BMI, family history, iodine levels (for thyroid hormones), menstrual cycle (for reproductive hormones), food consumption, smoking status and other drug use, medications, environmental stressors, and time of sampling; also, PBDEs should be adjusted for lipid content in biological samples due to their lipophilic nature. Furthermore, selective outcome reporting by many of the selected studies virtually limited the data available to conduct a meta-analysis aside from the low number of articles available. Additionally, the studies presented statistics of relationships in different forms that could not be reliably transformed into a single correlation coefficient value for comparison.

## 5. Conclusions

We consolidated and meticulously assessed the scientific literature regarding the relationship between PBDEs and the endocrine and reproductive system in this review, following the navigation guide methodology. Our findings suggest that PBDEs are positively associated with TT3, male testosterone, gestational age, birth length, and adverse birth outcomes and negatively associated with TT4, FT3, FT4, female FSH levels, neonatal head circumference, Apgar score, neonatal BMI, semen volume, sperm concentration, and total sperm count. However, some inconsistencies between the studies, such as marked selective outcome reporting, small sample sizes, and a lack of adequate confounder adjustments in the analyses, warrant further investigation to strengthen these findings.

With multiple in vitro and in silico studies demonstrating the endocrine-disrupting impacts of PBDEs and the selection of studies in this review identifying significant changes in hormone levels and their measurable effects on reproductive endpoints, it can be established with high confidence that PBDEs are endocrine-disrupting chemicals, even though the dose–response relationship of such effects within humans at all concentrations remain largely unknown due to the scarcity of large-scale observational studies that are designed considering a potential non-monotonic dose–response. To improve the quality of future studies, a consensus should be reached on the ideal methodological approach for observational studies regarding the hormonal and reproductive health effects of PBDEs. It is vital to comprehend this dose–response relationship to evaluate the potential threat to public health and structure policies that enable the effective implementation of health and safety measures.

## Figures and Tables

**Figure 1 ijerph-19-07820-f001:**
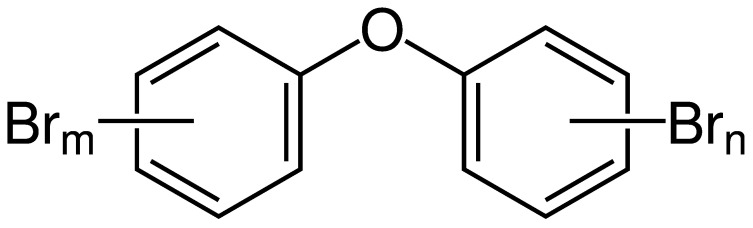
Structure of polybrominated diphenyl ethers.

**Figure 2 ijerph-19-07820-f002:**
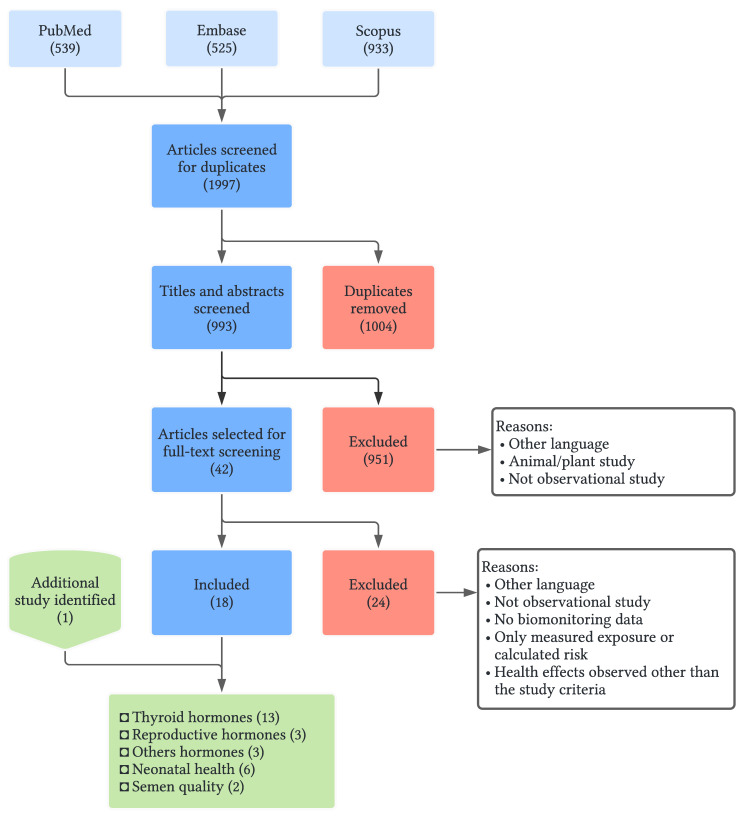
Literature search and screening flowchart.

**Table 1 ijerph-19-07820-t001:** Characteristics of the studies selected in the review.

Study Authors	Measurements and Samples Collected	Study Design	Participant Characteristics and Location	Comorbidities
Eguchi et al. [39]	Thyroid hormones and exposure levels through serum samples	Case–control	111-E: 77 workers at e-waste recycling site (Bui Dau, Vietnam), and C: 34 residents of unexposed rural area (Duong Quang, Vietnam)	None stated (participants with thyroid disease or taking thyroid-related medications were excluded)
Gravel et al. [40]	Thyroid and reproductive hormones and exposure levels through plasma samples	Cross-sectional	100-E1: 85 workers at e-waste recycling facility (Québec, Canada), and E2: 15 workers from recycling facilities other than e-waste (Québec, Canada)	None stated (participants with thyroid disease or taking thyroid-related medications were excluded)
Guo et al. [41]	Thyroid hormones and exposure levels through serum samples	Case–control	114-E: 57 sixth grade children in e-waste recycling town with >30 years of history (Q City, South China), and C: 57 individuals living 50 km from exposed town with no e-waste exposure (Q City, South China)	Respiratory illness symptoms
Guo et al. [42]	Thyroid hormones and exposure levels through blood samples	Case–control	112-E: 54 residents of e-waste recycling town with > 30 years of history (Q City, South China), and C: 58 individuals living 50 km from exposed town with no e-waste exposure (Q City, South China)	Neurological and respiratory illness symptoms
Julander et al. [43]	Thyroid hormones and exposure levels through serum samples	Longitudinal cohort study	14-E: 14 e-waste recycling workers (3 were excluded due to <4 sampling instances) (Örebro, Sweden)	None—one participant with thyroid disease excluded
Lv et al. [44]	Thyroid hormones and exposure levels through maternal serum, and neonatal health through physical examinations	Case–control	74-E: 64 pregnant women residing in e-waste exposed area for >5 years (Wenling, Taizhou, China), and C: 10 pregnant women residing for <2 years in non-e-waste exposed area (Wenling, Taizhou, China)	None stated
Wang et al. [45]	Thyroid hormones and exposure levels through plasma samples	Case–control	442-E1: 236 individuals occupationally exposed to e-waste (Taizhou, Zhejiang Province, China), E2: 89 individuals non-occupationally exposed (Taizhou, Zhejiang Province, China), and C: 117 individuals with no exposure to e-waste (Taizhou, Zhejiang Province, China)	None stated (participants with thyroid disease or taking thyroid-related medications were excluded)
Xu et al. [46]	Thyroid, ACTH, cortisol, and growth hormones and exposure levels through serum samples	Case–control	45-E: 21 residents of e-waste-exposed town with ≈20 years of history (Luqiao, Taizhou, China), and C: 24 individuals living 100 km from exposed site with no e-waste exposure (Tiantai, Taizhou, China)	None stated
Xu et al. [47]	Thyroid hormones (only FT3, FT4, and TSH) and exposure levels through serum samples	Case–control	55-E: 40 residents of e-waste-exposed region (Luqiao, Taizhou, China), and C: 15 individuals living 200 km from exposed town with no e-waste exposure (Yunhe, Taizhou, China)	None stated
Xu et al. [48]	Thyroid (only FT3, FT4, and TSH) and IGF-1 hormones and exposure levels through serum samples	Cross-sectional	167-E: 162 children in kindergarten in an e-waste-exposed town (Guiyu, Shantou, China)	None stated (participants with thyroid disease or taking thyroid-related medications were excluded)
Yu et al. [49]	Thyroid and reproductive hormones and exposure levels through serum samples, and semen quality through semen samples	Cross-sectional	76-E1: 38 men from e-waste area and living near e-waste dismantling plant (Qingyuan city, South China), and E2: 38 men living near an e-waste dismantling plant (Qingyuan city, South China)	None stated
Zheng et al. [50]	Thyroid hormones and exposure levels through serum samples	Cross-sectional	79-E: 79 e-waste recycling workers (Undescribed town, South China)	None stated (participants with thyroid disease or taking thyroid-related medications were excluded)
Zheng et al. [51]	Thyroid hormones and exposure levels through maternal serum samples, and neonatal health through physical examinations	Case–control	72-E: 48 residents of exposed area for >20 years (Wenling, Taizhou, China), and C: 24 residents of exposed area for < 3 years (Wenling, Taizhou, China)	None stated (participants with thyroid disease or taking thyroid-related medications were excluded)
Guo et al. [52]	Reproductive hormones and exposure levels through serum samples	Case–control	112-E: 54 residents of e-waste recycling town with >30 years of history (Q City, South China), and C: 58 individuals living 50 km from exposed town with no e-waste exposure (Q City, South China)	Neurological and respiratory illness symptoms
Yu et al. [53]	Semen quality and exposure levels through semen samples	Case–control	57-E: 32 residents of e-waste-exposed area with a history of decades (Longtang, Qingyuan, South China), and C: 25 semen samples from non-exposed population through semen bank (Qingyuan city, South China)	None stated
Li et al. [54]	Neonatal health through physical examinations, and exposure levels through umbilical cord tissue samples	Case–control	300-E: 150 women residing in e-waste-exposed area (Guiyu, Shantou, China), and C: 150 women residing in non-exposed area (Haojiang, Shantou, China)	None stated
Wu et al. [55]	Neonatal health through physical examinations, and exposure levels through umbilical cord serum samples	Case–control	167-E: 108 women residing in e-waste-exposed area (Guiyu, Shantou, China), and C: 59 women residing in non-exposed area (Chaonan, Shantou, China)	Upper respiratory tract infection and other diseases including anaemia, acute nephritis, skin disease, Pancreas Bile Syndrome, placental abruption, severe pregnancy-induced hypertension, preeclampsia, prolonged pregnancy, cord around neck, hepatitis A, and pregnancy-induced hypertension syndrome
Xu et al. [56]	Neonatal health through physical examinations, and exposure levels through placenta samples	Case–control	155-E: 69 women residing in e-waste-exposed area (Guiyu, Shantou, China), and C: 86 women residing in non-exposed area (Haojiang, Shantou, China)	None stated
Xu et al. [57]	IGF-1 hormone and exposure levels through umbilical cord serum samples, and neonatal health through physical examinations	Case–control	154-E: 101 women residing in e-waste-exposed area (Guiyu, Shantou, China), and C: 53 women residing in non-exposed area (Chaonan, Shantou, China)	None stated

Notes: E—exposed group; C—control group; ACTH—adrenocorticotropic hormone; FT3—free triiodothyronine; FT4—free thyroxine; TSH—thyroid-stimulating hormone; IGF-1—insulin-like growth factor 1.

**Table 2 ijerph-19-07820-t002:** Confounder adjustments and statistically significant findings (*p* < 0.05) of the studies selected in the review.

Study Authors	Confounder Adjustments	Significant Associations (*p* < 0.5)
Eguchi et al. [39]	Gender, age, BMI, perchlorate, iodide, thiocyanate, cholesterol, triglyceride, γ-GTP, living site, consumption of meat and eggs, and consumption of marine fish	None
Gravel et al. [40]	Age, BMI, blood cadmium and lead, and smoking status	TT4: ↑ BDE 209 (males); and FT3: ↓ BDE 153 and 209
Guo et al. [41]	Gender, BMI, and cough	None
Guo et al. [42]	Gender, BMI, dyspnoea, chest tightness, and smoking	TT3: ↑ BDE 47 and 99; TT4: ↓ BDE 153, 183 and ∑PBDEs; FT3: ↑ BDE 47; FT4: ↓BDE 153 and 183; and TSH: ↓ BDE 47 and 100
Julander et al. [43]	BMI	TT3: ↑ BDE 183; FT4: ↑BDE 28 and 100; TSH: ↑ BDE 99 and 154
Lv et al. [44]	Maternal age, pre-pregnancy BMI, gestational weeks, and maternal parity	None
Wang et al. [45]	Gender, age, plasma total lipids, alcohol consumption, and smoking status	None
Xu et al. [46]	None stated	ACTH: ↑ BDE 47, 99, 100, 153 and 154
Xu et al. [47]	Gender and age	FT4: ↓ BDE 47
Xu et al. [48]	Gender, age, and BMI	FT3: ↓ BDE 100 and ∑PBDEs; FT4: ↓ BDE 100, 153, 154 and 183; TSH: ↑ BDE 28, 47, 99, 100, 154, 209 and ∑PBDEs
Yu et al. [49]	Age, BMI, abstinence time, smoking status, and alcohol consumption	Serum-TT3: ↑ BDE 183; TT4: ↓ BDE 183; FT3: ↑ BDE 47; E2: ↓BDE 47 Semen-FT3: ↓; TSH: ↓ BDE: 154; total T: ↑ BDE 183; E2: ↓ BDE 154; semen volume: ↓ BDE 47 and 153; total sperm count: ↓ BDE 153
Zheng et al. [50]	Gender, age, BMI, and occupational exposure duration of e-waste recycling. ∑PBDEs were also adjusted for smoking.	TT3: ↑ BDE 47
Zheng et al. [51]	Maternal age, pre-pregnancy BMI, gestational weeks, and maternal parity	TT4: ↓ BDE 99 and 153
Guo et al. [52]	Females—dyspnoea, chest tightness, and palpitation Males—sore throat and loose cough	Males—Total T: ↑BDE 47, 100, 153, 183 and ∑PBDEs; Pr: ↑ BDE 47; LH: ↑ BDE 99, 100; E2: ↑ BDE 47, 209 and ∑PBDEs Females—FSH: ↓ BDE 153, 154, 183 and ∑PBDEs; LH: ↓ BDE 183; E2: ↑ BDE 153 and 183
Yu et al. [53]	Age, BMI, abstinence time, smoking status, and alcohol consumption	Sperm concentration: ↓ BDE 47; total sperm count: ↓ BDE 47
Li et al. [54]	None stated for Spearman’s correlation analysis (used for neonatal health outcome and PBDE concentration relationship analysis)	Head circumference: ↓ ∑PBDEs; birth length: ↑ ∑PBDEs; BMI: ↓ ∑PBDEs; Apgar score: ↓ ∑PBDEs
Wu et al. [55]	None stated	Gestational age: ↑ BDE 28; adverse birth outcomes: ↑ BDE 28, 47, 99, 153, 183, ∑PBDEs
Xu et al. [56]	Education, parity, smoking status, and alcohol consumption	Head circumference: ↓ BDE 47 and ∑PBDEs; birth length: ↑ BDE 47; BMI: ↓ BDE 47, 99, 100, 183, 209 and ∑PBDEs; Apgar score: ↓ BDE 28, 47, 153, 183 and ∑PBDEs
Xu et al. [57]	None stated	IGF-1: ↑ BDE 154 and 209; gestational age: ↑ BDE 100 and 154

Notes: BMI—body mass index; γ-GTP—gamma-glutamyl transpeptidase; TT3—total triiodothyronine; TT4—total thyroxine; FT3—free triiodothyronine; FT4—free thyroxine; TSH—thyroid-stimulating hormone; T—testosterone; Pr—progesterone; LH—luteinizing hormone; FSH—follicle-stimulating hormone; E2—estradiol; ACTH—adrenocorticotropic hormone; IGF-1—insulin-like growth factor.

**Table 3 ijerph-19-07820-t003:** Summary of risk of bias ratings assigned to the studies.

Domain	Eguchi et al. [39]	Gravel et al. [40]	Guo et al. [41]	Guo et al. [42]	Julander et al. [43]	Lv et al. [44]	Wang et al. [45]	Xu et al. [46]	Xu et al. [47]	Xu et al. [48]	Yu et al. [49]	Zheng et al. [50]	Zheng et al. [51]	Guo et al. [52]	Yu et al. [53]	Li et al. [54]	Wu et al. [55]	Xu et al. [56]	Xu et al. [57]
Study group representation																			
Knowledge of group assignments																			
Exposure assessment methods																			
Outcome assessment methods																			
Confounding																			
Incomplete outcome data																			
Selective outcome reporting																			
Financial conflict of interest																			
Other																			

Note: green—low risk of bias, blue—probably low risk of bias, amber—probably high risk of bias, and red—high risk of bias.

**Table 4 ijerph-19-07820-t004:** Summary of quality and strength assessment ratings across studies.

Criteria	Summary of Criteria	Thyroid Hormones	Reproductive Hormones	Other Hormones	Neonatal Health	Semen Quality
**Downgrade Criteria**						
Risk of bias	Study limitations—a substantial risk of bias across the body of evidence	0	0	−2	−1	−1
Indirectness	Evidence was not directly comparable to the question of interest (i.e., population, exposure, comparator, or outcome)	0	0	0	0	0
Inconsistency	Widely different estimates of effect in similar populations (heterogeneity or variability in results)	−1	0	0	0	0
Imprecision	Studies had few participants and few events (wide Cis as judged by reviewers)	−1	−1	0	0	−1
Publication bias	Studies missing from the body of evidence, resulting in an over or underestimation of the true effects from exposure	0	0	0	0	0
**Upgrade Criteria**						
Large magnitude of effect	Upgraded if modelling suggested confounding alone unlikely to explain associations with large effect estimate as judged by reviewers	0	0	0	0	0
Dose–response	Upgraded if consistent relationship between dose and response in one or multiple studies and/or dose–response across studies	0	0	0	0	0
Confounding minimizes effect	Upgraded if consideration of all plausible residual confounders or biases would underestimate the effect or suggest a spurious effect when results show no effect	0	0	0	0	0
**Overall quality of evidence**		Moderate	Moderate	Low	Moderate	Low
**Overall strength of evidence**		Limited evidence of toxicity	Limited evidence of toxicity	Inadequate evidence of toxicity	Limited evidence of toxicity	Limited evidence of toxicity

Note: the range of downgrade criteria is from 0 to −2, and the range of upgrade criteria is from 0 to 2, where 0 indicates no change to the rating.

## Data Availability

Not applicable.

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
