# Peer review of "Effects of Polybrominated Diphenyl Ethers on Hormonal and Reproductive Health in E-Waste-Exposed Population: A Systematic Review"

_ijerph, 2022, doi:10.3390/ijerph19137820_

Round 1

Reviewer 1 Report

Vishal Singh et al give an overview on the health effects of flame retardants in e-wastes in the manuscript titled “Effects of polybrominated diphenyl ethers on hormonal and reproductive health in e-waste exposed population: A systematic review

In my opinion the manuscript is well-written and logically built and processes the most current papers on this topic.

There are a few minor things, however that should be addressed before the paper could be accepted for publication.

Not everybody is familiar with polybrominated diphenyl ethers, therefore I recommend including a scheme showing the chemical structure of these compounds.

In the Introduction or Discussion sections a few sentences should discuss, the ways of exposure to these harmful compounds (dust, wastewater, skin contact). I think the harmful effect is largely correlated to the way of exposure.

As the authors correctly state the harmful effects cannot be unambiguously attributed to PBDEs, since e-wastes contain a number of other harmful compounds, such as lead. A few other toxic components should also be listed.

Author Response

Thank you for your kind suggestions and we have revised manuscript according your comments, please see attachment.

Reviewer 2 Report

Authors elaborated a relevant systematic review about the effects of PBDEs on hormonal and reproductive health in humans.

1.      General: Please consider to replace females with women and males with men.

2.      Lines 63-66. The sentence refers to octa- and penta-BDE commercial technical mixtures. Commercial decaBDE is restricted (http://chm.pops.int/Portals/0/download.aspx?d=UNEP-POPS-COP.8-SC-8-10.English.pdf).

3.      Line 116. Please, replace “comparator” with another term, ie “low-exposure group”, “reference”, or any other correct word in English.

4.      Lines 137-138. Is it necessary: VS, JCR and LMT?

5.      Table 3. Please, add legend of the colours. Table should be understood by itself.

6.      Table 4. Please, add legend of ratings. Table should be understood by itself. Upgrade Criteria share the same rating of 0. Thus, the upgrade criteria might be listed along with the rating of “0”as a footnote.

7.      Line 232. Please, replace “magnitudes” with “fold”.

8.      Line 210. Discussion. General. Please, consider to cite La Merril et al., 2019. Nat Rev Endocrinol 16, 45–57 (2020). https://doi.org/10.1038/s41574-019-0273-8. Consensus on the key characteristics of endocrine-disrupting chemicals as a basis for hazard identification.

9.      Lines 265-272. PBDEs are mixed type inducers of phase I and II enzymes. Such effect suggests a higher clearance of thyroid hormones, mechanism with collaborates with the disruption of the binding proteins. Such effect is also shown with PCBs. Please, add this additional mechanism.

10.   Lines 303-304. Please, consider to replace “this disparate result” with “these inconsistent results”.

Author Response

(The authors gave the same response as above.)
